



**Quantification of major particulate matter species from a single filter type using**
**infrared spectroscopy – Application to a large-scale monitoring network**
Bruno Debus[1], Andrew T. Weakley[1], Satoshi Takahama[2], Kathryn M. George[1,3], Bret Schichtel[4],
Scott Copeland[5], Anthony S. Wexler[1,6], Ann M. Dillner[1]*
[1] Air Quality Research Center, University of California, Davis, California, 95616, USA
[2] ENAC/IIE, Swiss Federal Institute of Technology Lausanne (EPFL), Lausanne, Switzerland
[3]Monitoring and Laboratory Division, California Air Resources Board, Sacramento, CA 95811,
USA
[4]National Park Service, Cooperative Institute for Research in the Atmosphere, Colorado State
University, Fort Collins, CO 80523, USA
[5] Cooperative Institute for Research in the Atmosphere, Colorado State University, Fort Collins,
CO, 80523, USA
[6]Departments of Mechanical and Aerospace Engineering, Civil and Environmental Engineering,
and Land, Air and Water Resources, University of California, Davis, California, 85616, USA
*Correspondence to:* Ann M. Dillner (amdillner@ucdavis.edu)
**Abstract**
To enable chemical speciation, monitoring networks collect particulate matter (PM) on different
filter media, each subjected to one or more analytical techniques to quantify PM composition
present in the atmosphere. In this work, we propose an alternate approach that uses one filter
type (teflon or polytetrafluoroethylene, PTFE, commonly used for aerosol sampling) and one
analytical method, Fourier Transform Infrared (FT-IR) spectroscopy to measure almost all of the
major constituents in the aerosol. In the proposed method, measurements using the typical
multi-filter, multi-analytical techniques are retained at a limited number of sites and used as
calibration standards while sampling on PTFE and analysis by FT-IR is solely performed at the
remaining locations. This method takes advantage of the sensitivity on the mid-IR domain to
various organic and inorganic functional groups and offers a fast and inexpensive way of
exploring sample composition. As a proof of concept, multiple years of samples collected within
the Interagency Monitoring of PROtected Visual Environment network (IMPROVE) are explored
with the aim of retaining high quality predictions for a broad range of atmospheric compounds
including total mass, organic (OC), elemental (EC) and total (TC) carbon, sulfate, nitrate and



31 crustal elements. Findings suggest that models based on only 21 sites, covering spatial and
32 seasonal trends in atmospheric composition, are stable over a three year period within the
33 IMPROVE network with prediction accuracy ($R^2 > 0.9$, median bias less than 3% for most
34 constituents. Incorporating additional sites at low cost or partially replacing existing, more time
35 and cost intensive techniques are among the potential benefits of one-filter, one-method
36 approach.



## 1 Introduction

In the United States, filter-based chemical speciation of ambient aerosols has been in operation for decades to quantify trends, assess transport and atmospheric transformation, identify sources of air pollution, evaluate impacts of pollution regulations, assess impacts on visibility, radiative forcing, human and ecosystem health and evaluate atmospheric and climatological models. The two federally funded speciation networks, the Interagency Monitoring of PROtected Visual Environments (IMPROVE) and the Chemical Speciation Network (CSN) collect 24-hour filter samples using three filter media: polytetrafluoroethylene for analysis by gravimetry, hybrid integrating plate and sphere (HIPS), and x-ray fluorescence (XRF), quartz for thermal optical reflectance (TOR) and nylon for ion chromatography. Over the decades of operation, the analytical methods have evolved with efforts to maintain consistency in trends while also adopting improved methodology and retiring obsolete equipment. Impacts of many of these changes have been addressed in the literature (Hyslop et al., 2015, 2012; White et al., 2016; Zhang et al., 2021; Chow et al., 2007a, 2015) and in data advisories posted on the IMPROVE website (http://vista.cira.colostate.edu/Improve/data-advisories/).

In this paper, we explore the use of Fourier transform-infrared spectroscopy (FT-IR) to reproduce most of the existing speciation data because the most components exhibit optical activity in the mid-IR. The number and bands of organic compounds are numerous, but generally group frequencies can be found above 1500 cm$^{-1}$ and compound-specific spectral patterns ("fingerprint region") below this frequency; down to approximately 700 cm$^{-1}$ (for example,(Weakley et al., 2016; Mayo et al., 2004). Graphitic carbon displays peaks near 1600 cm$^{-1}$ due to lattice defects (Tuinstra and Koenig, 1970; Friedel and Carlson, 1971), displacement vibrations near 868 cm$^{-1}$ (Nemanich et al., 1977), and a broad, sloping absorbance between 4000 and 1500 cm$^{-1}$ due to the tail of the electronic transition more strongly observed in the UV (Parks et al., 2021). Inorganic substances containing polyatomic ions such as sulfate, nitrate, and ammonium have strong vibrational modes above 600 cm$^{-1}$ (Mayo, 2004). Crystalline and amorphous geological minerals in the form of oxides (which include hydroxides and oxyhydroxides) have distinct internal vibrational modes influenced by the electronegativity of the metal to which the oxygen is bonded (Busca and Resini, 2006; Chukanov and Chervonnyi, 2016; Margenot et al., 2017).

FT-IR spectra with partial least squares (PLS) calibrations have been shown to reproduce OC and EC concentrations using organic and graphitic carbon absorption bands, respectively, at a limited number of sites in the IMPROVE network (Dillner and Takahama, 2015a, b; Reggente et al., 2016), CSN (Weakley et al., 2016, 2018a) and FRM (Weakley et al., 2018b). Takahama et al. (2019) reviews these findings and the overall framework to be used for the two phases of such statistical calibrations: model building (sample selection, spectral preparation, model generation, model selection, model evaluation, and model understanding) and operation (error anticipation and model updating). Inorganic ions and geological mineral absorption bands have been used for



chemical speciation of these substances using FT-IR in prior studies (e.g., Cunningham et al.,
1974; McClenny et al., 1985; Pollard et al., 1990; Bogard et al., 1982; Foster and Walker, 1984).
Organic absorption bands are useful for measuring OC but also provide spectral information
needed to add detailed knowledge of composition not currently measured in air quality
monitoring networks – such as organic matter (OM) and organic functional group composition –
which is the subject of other work (Reggente et al., 2019; Boris et al., 2019, 2021; Burki et al.,
2020). Such calibrations, also combined with factor analytic approaches, can provide source
characterization on par with more costly mass spectrometric techniques (Boris et al., 2021;
Yazdani et al., 2021a; Hawkins et al., 2010; Takahama et al., 2011; Liu et al., 2012; Corrigan et al.,
83 2013).

Although FT-IR shows promise for measuring many constituents in aerosol, it is not without its
challenges. One limitation is that not all PM constituents can be measured, or measured with
high sensitivity, from the FT-IR spectrum. For instance, NaCl and $MgCl_2$ do not have IR-active
substituents. While a multitude of spectral signatures associated with mineral dust arise from
their constituent bonds – e.g., the metal-oxygen bonds in oxides (the oxide form is explicitly
assumed in estimating dust mass concentrations from elemental composition for the IMPROVE
network), some must be predicted from correlation with other constituents (e.g., some forms of
iron) if IR-activity is lacking. Other substances are IR-active but have weak responses, such as
graphitic carbon (Niyogi et al., 2006; Parks et al., 2021). The absorption and scattering by the
PTFE filter also pose challenges for quantitative analysis. The PTFE-based material changes over
time due to change in manufacturer or manufacturing process, and is difficult to fully characterize
a priori or treat with simple blank subtraction techniques. PTFE absorption limits full access to
the range of spectroscopic information in the mid-IR, for instance in the region of carbon-oxygen
bonds that can lead to less than full recovery of OM mass. Additionally, scattering leads to
broadly-varying slope in the group frequency region. This scattering artifact is minimized by
baselining (Kuzmiakova et al., 2016) and using many standards that have a range of scattering
and absorption observed in the network (Debus et al., 2019), yet these techniques can still lead
to errors in quantification. Weakley et al., (2018b) demonstrated that calibrations built using one
brand of filter can be accurately extended to another brand of PTFE filter with numerically
marginal but statistically significant increase in method error (e.g., +2% error for α=0.05).
However, these studies are insufficient to generalize findings to all types of sampling filters.
The goal of this work is to assess the capability of using FT-IR to measure the aerosol chemical
composition at all IMPROVE sites. FT-IR quickly and non-destructively collects information-rich
spectra from routinely collected PTFE filter samples. Ambient samples from strategically-
selected IMPROVE sites are used for calibration and reasonably mimic the composition, matrix
effects and substrates of the unknowns, all of which theoretically lead to accurate estimations of
concentrations. Using all samples from selected sites reduces maintenance, shipping, processing



and coordinating required to maintain intermittent quartz and nylon filter sampling at every site.
Sites are selected using data from 2015 and are used for calibrating samples from 2015-2017.
Samples from all other (non-calibration) IMPROVE sites are predicted and compared to routine
IMPROVE data to assess the quality of prediction. Aerosol components to be measured include
TC, OC, EC, inorganic ions, soil elements, particulate mass, and light absorption.

## 2   Methods

### 2.1   Network data

IMPROVE samples collected every third day at all North American sites (Section S1) from January
2015 through December 2017 are included in this study. Fine particulate matter (aerodynamic
diameter less than 2.5 micrometers) is deposited on 25 mm diameter filters
polytetrafluoroethylene (PTFE) and quartz filters by sampling air at a nominal flowrate of 22.8
liters per minute from midnight to midnight local time. Parallel 37 mm nylon filters are collected
at the same flow flowrate. PTFE filters are analyzed by multiple instruments and archived for
future analysis. Nylon filters and a portion of each quartz filter undergoes destructive analysis
and a remaining part of the quartz filters are retained for archive.
Over the period covered in this study, two different TOR instruments were employed to measure
OC, EC and TC. Quartz filters sampled prior to 2016 where analyzed on a DRI Model 2001 thermal
optical carbon analyzers (Chow et al., 1993) while filters collected beginning in January of 2016
were analyzed on a DRI Model 2015 multi-wavelength thermal optical carbon instrument (Magee
Scientific – Berkley, USA)(Chow et al., 2015). Both instruments use the IMPROVE_A protocol
(Chow et al., 2007b), which outlines the temperature step, gaseous environment in the
instrument and that reflectance is used to define the split point between OC and EC. To correct
for gas phase adsorption onto the quartz filter, the monthly median field blank OC concentration
is subtracted from each OC measurement during that sample month. Carbon concentrations are
reported in $\mu g/m^3$.
An in house Hybrid Integrating Plate and Sphere (HIPS) system evaluates light absorption from
the PTFE filters in the IMPROVE network (White et al., 2016). In this work, the measured
absorption coefficient (*Fabs*) is converted into a TOR EC equivalent concentration assuming a
Fabs / EC ratio of 10 $m^2g^{-1}$ (Malm et al., 1994). The resulting value, referred to as HIPS Black
Carbon (HIPS BC), is used as part of a quality control procedure to evaluate potential outliers in
TOR EC measurements.
Data from gravimetry and X-ray fluorescence (XRF) analysis obtained from PTFE filters and ion
chromatography from the nylon filters are also used in this study. Additional information on
routine IMPROVE methods can be found on the IMPROVE website
(http://vista.cira.colostate.edu/Improve/). IMPROVE data are available online at
(http://views.cira.colostate.edu/fed).



## 2.2    Outlier removal


Data were screened for outliers to eliminate their influence on the results.  Out of the ~61,500
total number of samples in the three-year period, fewer than 800 were excluded from the
analysis due sampling issues or missing TOR, XRF or IC data. In addition, 65 samples collected at
the Wheeler Peak Wilderness (New Mexico) site between November 2015 and April 2016 were
excluded due to an EC contamination caused by a diesel-powered ski lift.
Potential outliers in TOR measurements were investigated by regressing TOR EC against HIPS BC
concentrations. Samples with differences exceeding a predefined threshold value (0.68 $\mu g/m^3$)
were flagged as potential outliers (section S2). The status of these samples was confirmed by
building separate TOR EC and HIPS BC calibrations. The poor agreement between TOR EC and FT-
IR EC concentrations contrasts with the nearly 1:1 relationship HIPS BC and FT-IR BC predicted
values indicating that TOR EC concentrations were likely compromised (Section S2). For the
period considered in this study, 112 samples with faulty TOR EC values were identified and
excluded from further analysis. The number of valid sample spectra retained in this study is
161    61,462.

## 2.3    Fourier-transform infrared (FT-IR) analyses

Since 2015, all PTFE in the IMPROVE network have been analyzed by infrared spectroscopy for
research and evaluation purposes. FT-IR measurement occurs after gravimetric analysis and prior
to XRF and HIPS to prevent possible loss of volatile species under the mild vacuum in XRF. Three
FT-IR spectrometers including one Tensor 27 and two Tensor 2 instruments (Bruker Optics,
Billerica, MA) equipped with a pre-aligned mid-IR source and a liquid nitrogen-cooled wide-band
mercury cadmium telluride (MCT) detector were used for spectra acquisition in the range 4000 -
420 $cm^{-1}$ by averaging 512 scans at a nominal resolution of 4 $cm^{-1}$. The single beam signal
associated with each PTFE filter was converted to an absorbance spectrum using the most recent
zero reference signal, updated hourly.
Previously, it was determined that calibration transfer between multiple FT-IR instruments is not
required as long as their spectral response is carefully matched by controlling a set of key
environmental and instrumental parameters (Debus et al., 2019). Briefly, each mercury cadmium
telluride (MCT) detector is connected to an automatic liquid nitrogen micro dosing system
(NORHOF, Ede, Netherlands) designed to improve signal stability and maintain a high signal to
noise ratio. The repeatability and reproducibility of the filter position relative to the IR beam is
controlled via a house-built sample chamber (4.0 × 5.1 × 4.5 cm) mounted inside the instrument
sample compartment. Details regarding the chamber design have been published elsewhere
(Debus et al., 2019). Finally, the contribution of water vapor and carbon dioxide to the signal was
minimized by continuously purging both the sample chamber and the optical bench with a VCD
Series $CO_2$ adsorber / dryer system (PureGas LLC, Broomfield, CO). For each sample, the
acquisition procedure involves a 4 minutes purge period followed by a spectrum collection lasting



about 1 minute. An in-house macro interfaced to the OPUS software (Bruker Optics, Billerica,
MA) controls each step. PTFE filters were measured in transmission mode without sample
preparation.  No interpolated data (from zero-filling) are included in the final raw spectra.
Collected spectra are subjected to weekly quality control procedures detailed in (Debus et al.,
2019). Duplicate and replicate measurements were also performed to evaluate instrument
stability and found to be within +/- 10%.
## 2.4    Multivariate Calibration Model - Partial Least Squares (PLS) Regression
While the presence of certain category of atmospheric compounds can be identified qualitatively
from an FT-IR spectrum, an accurate quantification of their concentration requires calibration.
PLS is a commonly used algorithm to relate a multi-wavenumber measurement to any particular
sample properties such as concentration (Wold et al., 2001). In brief, PLS maximizes the co-
variance between a set of response variables (species measurements) and a reference
measurement (FT-IR spectra) from which equivalent predicted values are desired. In so doing,
the optimal combination of response variables best describing the reference measurement is
identified and the selected features are used to build a multivariate calibration.  With all least-
squares calibration methodologies, concentration-dependent biases in residuals that are
determined by the quality of fit ($R^2$) and dynamic range of the data are expected due to the nature
of least-squares estimation (Besalú et al., 2006; Draper and Smith, 1998, pp. 63-64,173,638).  For
further discussion of these biases, see Section S1.
The applicability of PLS to quantify carbonaceous aerosol species (Reggente et al., 2016; Weakley
et al., 2016, 2018a) or functional groups (Boris et al., 2019; Ruthenburg et al., 2014) collected on
PTFE filters in various monitoring networks and field campaigns has been successfully
demonstrated. A complete review of the implementation of PLSR calibration in the framework
of atmospheric particulate matter characterization has been recently published (Takahama et al.,
208    2019).

To evaluate model performance, FT-IR predicted concentrations were regressed against their
reference measurement to quantify residuals and a series of metrics. Reported figures of merit
include the coefficient of determination ($R^2$), bias, error and the method detection limit (MDL).
Residuals are defined as the difference between predicted and reference concentrations, bias
corresponds to the median residual while error is the median absolute residual. To facilitate inter-
model comparison, relative performance metrics were calculated by normalizing the values by
their reference value. FT-IR PLSR calibration MDL was estimated from field blank predicted
concentrations as the 95[th] percentile minus the median residuals, as is done for other species in
the        IMPROVE        network        http://vista.cira.colostate.edu/improve/wp-
content/uploads/2021/07/IMPROVE-SOP-351_Data-Processing-and-Validation_2021_final.pdf.
Performance is reported for all samples together regardless if the samples were included in the
calibration. This enables comparison between models with different samples used for calibration.



For further insight into model prediction accuracy, the distribution in FT-IR residuals is
qualitatively compared with residuals from collocated measurements. Collocated quartz filters
are collected at the Everglades (FL), Hercules-Glades (MO), Medicine Lake (MT) and Phoenix (AZ)
sites. Similarly, collocated Teflon filters are sampled at Mesa Verde (CO), Proctor Maple Research
Facility (VT), Saint Marks National Wildlife Refuge (FL), Yosemite (CA) and Phoenix (AZ) sites while
collocated nylon filters are featured at the Phoenix (AZ), Frostburg Reservoir (MD), Mammoth
Cave (KY) and San Gabriel (CA) sites.
Data handling and analysis was performed in Matlab R2018a (The MatWorks, Inc, Natick, MA,
United States) using the statistics and signal processing toolboxes. PLS was computed via the
libPLS Matlab package (v1.9) (Li et al., 2018).

### 2.5    FT-IR Calibrations for Predicting PM Composition

This section presents the design of calibrations for quantifying the concentration of major
atmospheric species by taking advantage of the composition-based information embedded
within an FT-IR spectrum. In practice, spectra are calibrated against reference measurements
from TOR, XRF, IC, HIPS and gravimetric analysis with the aim of predicting concentrations of
atmospheric constituents using only spectra of PTFE filters as input.
A multilevel model (Snijders and Bosker, 2011; Takahama et al., 2019) is proposed in which
dedicated calibration models for subgroups of samples are constructed, and applied according
to a predetermined selection criterion for each sample. This model considers two subgroups: i)
samples determined to be dominated by biomass burning, which are calibrated with similar
samples, and ii) the remaining samples, which are calibrated with samples from a limited number
of sites. To establish baseline performance metrics for comparison, a "Global model" in which a
single calibration (for each species) is constructed from all samples considered together and
described in Section S1 (Supplement).
The first step in the development of the Multilevel model consists of screening for biomass
burning samples. These samples are removed from consideration during the site selection
process.  A simple detection method combining estimates of key functional group spectral peak
areas and spectral dissimilarity metrics were used to segregate biomass burning samples from all
other samples. Next, a Gaussian Mixture Model (GMM) was applied to the spectra of all non-
biomass burning samples.  The GMM exploits the specificity of the infrared signal for organic and
inorganic species.  The GMM was implemented with the aim of clustering the non-biomass
burning FT-IR spectra into groups sharing similar spectral features (Section 2.5.2). Those groups
were later used as part of the methodology for selecting sites with representative atmospheric
composition. Spectra from the year 2015 were used as a benchmark to validate the biomass
burning detection strategy, build the GMM and establish the list of representative sites where



multi-filter collection/multi-analyses should be retained (section 2.5.2). The identified biomass
burning samples are used to build a calibration for biomass burning samples (Section 2.5.1).
### 2.5.1   Biomass burning model
FT-IR spectra were used to estimate functional group areas and calculate spectral dissimilarities
metrics to segregate biomass burning samples from all other samples. Although this paper
focuses on using FT-IR to measure the major aerosol components in routine speciated aerosol
monitoring networks, FT-IR is more frequently used to measure organic functional groups (e.g.
(Russell et al., 2011; Ruthenburg et al., 2014; Boris et al., 2019). Specific regions in the IR spectra
correspond to specific functional groups. Peak areas, calculated from baseline corrected spectra
(see Section S3 for baseline procedure), for carbonyl, OH and CH were used rather than functional
group calibrations for simplicity. Because the relative functional group peak area tends to
increase significantly as the cumulative peak area decreases, typically for low mass deposition
samples, an estimate of spectral dissimilarities, the squared Mahalanobis distance ($D_i^2$), for each
site is also considered to minimize false detection. The Mahalanobis distance (Mahalanobis,
1936; Cios et al., 1998) is a measure of the spectral dissimilarity between a given spectrum at a
site and the mean spectrum at the site. Taking advantage of $D_i^2$ and relative functional group
areas, a set of criteria were established from observations at known wildfire sites during wildfire
season (O'Dell et al., 2019). First, samples collected under heavy smoke conditions whose spectra
fulfill C–H ≥ 2 %, C=O ≥ 15 % and $D_i^2 \geq 3\,\overline{D^2}$ were detected (Section S3). This group of spectra
tend to have large $D_i^2$ values and, consequently, the $3\,\overline{D^2}$ threshold often excludes samples with
low to moderate biomass burning contributions. For a more inclusive detection, spectra from the
first group were removed from consideration, the $D_i^2$ values are updated for each sample and
the plots were regenerated. The cut-off value for the relative carbonyl functional group area was
lowered to 8 % while other parameters were not changed. Spectra identified by the first and
second rounds are considered biomass burning samples. This procedure is performed for each
site and for each year of sample collection (Section S.3)
Recent work has shown that smoke samples may be identified using techniques such as cluster
analysis and labeling (Burki et al., 2020) similar to the GMM used here and through detection of
molecular markers – levoglucosan and lignin – or peak profiles in FT-IR spectra (Yazdani et al.,
2021a, b). For the large data set in this work (~20,000 samples in 2015), cluster analysis
resulted in multiple clusters that could be associated with smoke-impacted samples likely due
to the variations in fuel, oxidation conditions, and contributions from other sources. Therefore,
for this work we selected a single group of smoke-impacted samples based on specific organic
features known to be present in FT-IR spectra. While the criteria for smoke-impact labeling can
be defined differently according to each intended purpose, the method presented here is
demonstrated to sufficiently partition the samples for building accurate submodels to predict
concentrations of PM constituents.





While ions and crustal species are not necessary correlated with wildfire emissions, the Biomass
Burning sub-model for accounts for interferences that are necessary to track in order to maintain
high prediction accuracy for samples collected on smoky days.
2.5.2    Limited Sites Model
To assess major PM$_{2.5}$ composition regimes in the network and to identify representative sites to
use as calibration standards in the Limited Sites model, screening of all FT-IR spectra (except
samples identified as biomass burning samples) across all locations and seasons was performed
by building a Gaussian Mixture Model (GMM) (Bilmes, J. A., 1998; Hastie, T et al., 2009). The basic
idea behind GMM is to group FT-IR spectra into clusters of similar spectral shape using a
probabilistic approach describing the likelihood that any given spectrum belongs to a particular
class. To minimize the concentration dependence and emphasize composition, raw spectra were
transformed to second derivative spectra using a 2$^{nd}$ order, 21 point, Savitzky-Golay filter
(Savitzky and Golay, 1964), differenced with filter blank spectra and divided by their respective
Euclidean norm (Bro and Smilde, 2003). Additional details about the GMM pre-processing and
implementation as well as cluster interpretation are provided in Section S4.
After classification, a single site per cluster was selected to represent the atmospheric
composition captured in that cluster. For any given cluster, the retained location was defined as
the site with the largest number of classified spectra with the highest probabilities of belonging
to that cluster. To prevent misleading site selection and enhance spatial coverage, the following
set of decision rules were used: *i)* if the same site is representative of two clusters, it is ascribed
to the cluster with the largest number of classified spectra from that site, *ii)* if none of the
retained sites accounts for a given spatial region or known source type in the network, an
additional site with the highest number of classified spectra is selected from a nearby cluster,
and *iii)* only sites under continuous operation between 2015 and 2017 are eligible for selection.
Criteria *ii)* was invoked once to add a site in the Midwest to improve spatial coverage in that
region and to capture prescribed fire emissions in Kansas.  All non-biomass burning samples from
selected sites were used as FT-IR calibration standards for all species and all non-biomass burning
samples are predicted with these models.  Once established, the selected sites are not re-
evaluated but instead were used in all subsequent years as would occur in practice.
2.5.3    Application of Multilevel Model
Multilevel model is the combined FT-IR predicted concentrations from the Limited sites and
Biomass burning models.  Multilevel modeling will be discussed in the context of carbonaceous
aerosols before extending the modeling to other atmospheric constituents with detectable
infrared signatures. In addition to OC and EC, species evaluated for FT-IR prediction include PM$_{2.5}$
mass, soil elements (silica, aluminum, calcium, titanium, iron), anions (sulfate, nitrate) and HIPS
BC. Next, the years 2016 and 2017 will be examined to assess the long-term stability of the





proposed Multilevel strategy by screening for smoke samples and re-calibrating each year using
the sites selected using 2015 data.

## 3   Results and discussion

In the following sections, the quality of FT-IR based calibrations for quantifying aerosol
composition across continental US and their long-term applicability to large speciation
monitoring networks will be assessed. Section 3.1 describes the selected calibration samples for
the Biomass Burning and Limited Sites models.  In Section 3.2, Biomass Burning and Limited
model performance will be briefly reviewed before exploring the Multilevel FT-IR predictions for
all samples. Initially focused on carbonaceous species on PTFE samples collected in 2015, FT-IR
predictions will be extended to other atmospheric constituents and years.

### 3.1    Multilevel modeling – Calibration sample selection

#### 3.1.1  Biomass burning sample selection

Using the methods described above, 492 samples impacted by biomass burning emissions were
identified in 2015 (2.5 % of the network), 288 samples in 2016 (1.5 %), and 817 samples in 2017
(3.7 %).  The mean OC concentration of the biomass burning samples range was 5.6 – 8.3 µg/m$^3$
with maximum concentrations extending from 44.5 to 97 µg/m$^3$ over the three year period.
Similarly, per year, the mean EC concentration varies between 0.61 – 0.9 µg/m$^3$ with maximums
up to 2.9 – 3.9 µg/m$^3$.  Mean OC/EC ratios are larger than 7, in agreement with past literature
(Schichtel et al., 2008; Sorooshian et al., 2011). Analysis of the detected samples shows reliable
spatial and seasonal distributions, consistent with biomass burning emissions predominantly in
summer and fall across the Pacific North West and Northwestern US (Section S3).  Two-thirds of
the identified samples were selected (Section S5) as calibration standards for the calibration and
resulting model was applied to the remaining third of the smoke impacted samples.

#### 3.1.2  Limited Sites model – clusters and retained sites

Figure 1 shows the spatial distribution of the 21 sites selected for Limited sites model. From a
spatial standpoint, retained sites appear reasonably scattered across the network including
Hawaii and the Virgin Islands.  Clusters are represented by a distribution of urban and rural sites.
One urban cluster is represented by Fresno and contains mostly urban samples from Fresno and
Phoenix. All other clusters contain mostly rural and pristine sites.  However, two other urban sites
were retained, Phoenix and Birmingham.  The Phoenix cluster contains samples from the
southwest in the spring.  The Birmingham site along with the Tallgrass site represent a non-
western cluster in the spring and summer.

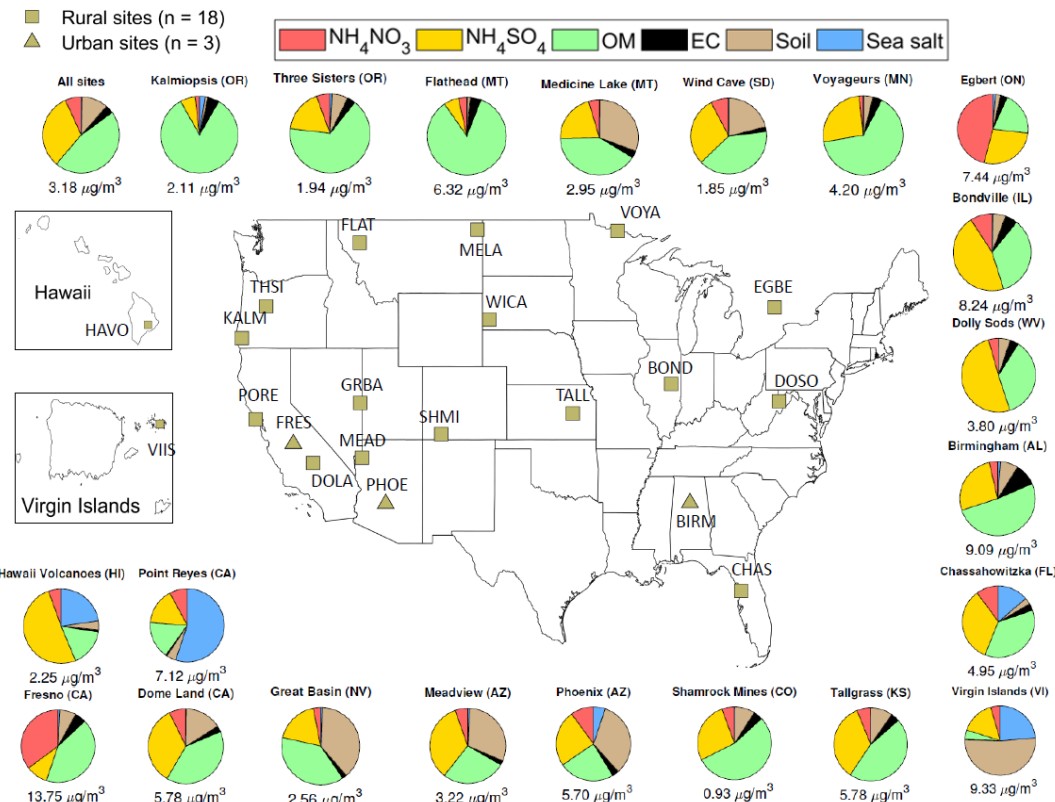

**Figure 1:** Spatial distribution, median PM₂.₅ concentration and composition of the 21 representative sites. Sites are identified by the four letter site code which is the first four letters of a single word site name (Fresno = FRES) or the first two letters of the first and second word for two word site names (Dome Land = DOLA. The composition is obtained from IMPROVE measurements and the IMPROVE reconstructed fine mass equation. The top left pie chart representing the median PM₂.₅ composition across all sites and samples is given for comparison.

The clusters are also seasonally distributed (Section S6): five clusters are dominated by fall - winter samples, ten clusters containing summer samples (along with varying number of spring and fall samples), two clusters are predominately spring and one is spring - fall. Three clusters have little seasonality.

Because FT-IR spectra are clustered based on composition, the first step in assessing the representativeness of the 21 sites is to compare the concentration ranges. For this purpose,

distributions in TOR OC and EC concentrations excluding biomass burning samples are compared
for the 21 sites used for calibration and the 140 remaining sites. In Fig. 2, the two probability
density functions are very similar for both OC and EC despite large differences in sample
populations (2572 and 16,543, respectively). In addition to matching the range of carbonaceous
concentrations observed in the rest of the network, the presence of species interfering with
organic functional groups should also be accounted for by the calibration. Because ammonium
absorptions overlap with carbonaceous absorptions, situations where ammonium to OC and
ammonium to EC ratios are different between calibration samples and samples to be predicted
were associated with additional sources of bias and error (Dillner and Takahama, 2015a, b).
Although not measured in IMPROVE, ammonium concentration is approximated from nitrate and
sulfate assuming both species are fully neutralized. The corresponding probability distribution in
Fig. 2 confirms the equivalence between the ranges of ammonium/OC and ammonium/EC
concentrations spanned by the Limited sites samples and the overall network. In section S6, Fig.
S6-3 shows reasonable agreement between the selected sites and the rest of the network for
PM2.5 mass, ions, elements and HIPS BC.  Together, these results suggest the list of 21 sites is a
suitable representation of network variations in OC and EC and their relative proportion to
ammonium, and for all other predicted constituents.

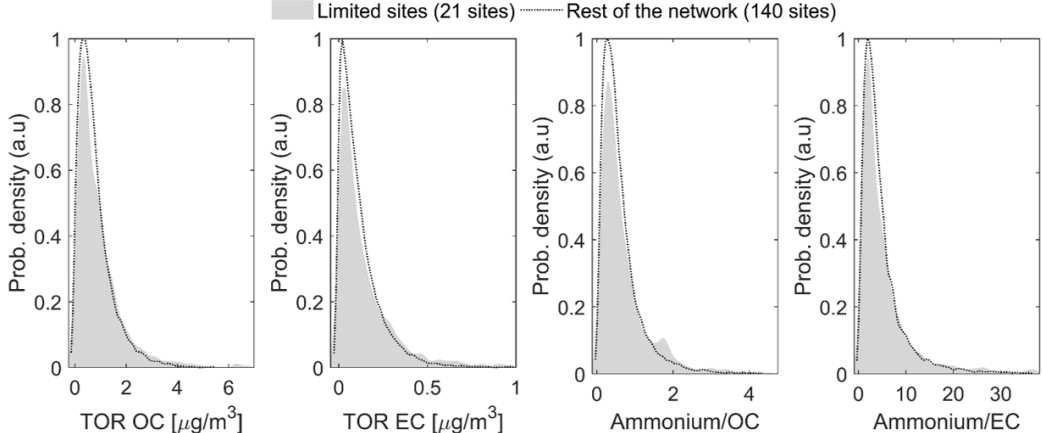

**Figure 2**: Comparison of probability density function for OC, EC and ammonium concentrations
in 2015 between the 21 sites retained for Limited calibration and the rest of the network.
The spatial and seasonal as well as the urban and rural diversity supports the compositional
diversity of the selected sites as shown in Fig. 1.  The three urban sites have distinct
characteristics.  At the Fresno, CA site, the composition is dominated by nitrate (35 %) and organic
matter (42 %) with an autumn – winter pattern consistent with agriculture and residential wood
burning activities (Ngo et al., 2010) as well as with the formation of secondary aerosols during



stagnation events and a low inversion layer (Watson and Chow, 2002). Phoenix, AZ site features
a strong soil component (33%) associated with spring dust storms and windblown dust and equal
proportions of ammonium sulfate (25 %) and OM (24 %) occurring mostly in spring and summer.
The ammonium sulfate and organic matter has been attributed to regional power production and
traffic (Brown et al. 2007). In contrast, Birmingham samples show little seasonal trend with
elevated OM (52 %) and EC (10 %) fractions originating from various combustion processes
including vehicle exhaust, biomass burning and biogenic secondary organic aerosols (Blanchard
et al. 2016). The other dominant species at this site is ammonium sulfate (26 %), characteristic of
coal burning and industrial activities in the East (Watson et al. 2015).
Among rural sites, four noticeable patterns in PM$_{2.5}$ composition are distinguishable. The first
corresponds to OM fractions accounting for more than two-thirds of the filter mass. High OM
samples are encountered at four locations in Northwestern US, namely the Kalmiopsis (OR),
Three Sisters (OR), Flathead (MT) and Voyageurs (MT) sites. Samples from Voyageurs (MN) and
Flathead (MT) sites are from Summer-Fall and present elevated median PM$_{2.5}$ concentrations
(4.20 µg/m$^3$ and 6.32 µg/m$^3$, respectively) and very large percentage of OM consistent with
biomass burning emissions. Kalmiopsis (OR) and Three Sisters (OR) samples have a lower and
nearly identical median PM$_{2.5}$ concentration ($\approx$ 2 µg/m$^3$) but differ in their monthly distribution
with the former displaying more winter samples than any other season whereas the later shows
little seasonality.
The second type of sites have high OM and sulfate concentrations. Both Shamrock Mines (CO)
and Tallgrass Prairie (KS) sites have larger OM than sulfate content. However, the Colorado site
has more autumn – winter samples, represents samples in the Rockies and Alaska and an overall
small median PM$_{2.5}$ concentration (< 1 µg/m$^3$). The Kansas site has a majority of spring samples,
representing non-western samples and has a significantly larger PM$_{2.5}$ concentration ($\approx$ 6 µg/m$^3$)
that is attributed to prescribed burning near the Tallgrass site (Whitehill et al. 2019).   Other sites
have higher median sulfate concentrations (~50%) than OM concentrations (~40%) such as Dolly
Sods (WV) and Bondville (IL). The monthly sample distribution indicates seasonal influences:
Bondville (IL) samples are mostly from the summer and the concentrations are relatively high
while the Dolly Sods (WV) site samples are mostly not in the summer with lower concentrations.
Because the spectra were normalized to minimize influence of concentration, these two clusters
likely have different organic composition even though the relative amount of OM is the same.
Finally, situations where sulfate and OM are present in equal proportions ($\approx$ 36 %) are reported
at the Dome Land (CA) and Chassahowitzka (FL) pristine sites mainly featuring spring – summer
and winter samples, respectively.
A third group of noteworthy PM$_{2.5}$ compositions at rural sites contain a large fraction of (> 20 %
of the total mass). The Virgin Islands (VI) site presents the highest soil fraction across the network
52 % of the total PM$_{2.5}$ mass, mostly originating from long-range Sahara soil dust transport





(Holmes and Miller 2004).  In addition to sulfate and OM, elevated soil contributions are observed
for the Wind Cave (SD), Meadview (AZ), Medicine Lake (MT), and Great Basin (NV) sites with soil
content between 20 and 40%.  Although the seasonality is somewhat different between these
sites, they all have many samples from the spring suggesting the dust is due at least in part to
spring dust storms and may also contain resuspended road dust and more localized dust sources.
A fourth and final distinct category of PM$_{2.5}$ composition includes a collection of sites with unique
local atmospheric pollution sources, specific to those locations. The Hawaii Volcanoes (HI) site
where sulfur emitted as part as the volcanic activity, contains 51% sulfate along with sea salt (23
%). Another location with unique composition is the Point Reyes (CA) site where the sea salt
contribution reaches 55% of the median filter mass for the clustered samples, larger than any
other marine site in the network. Finally, the Egbert (ON) Canadian site, representing the upper
Midwest in winter is dominated by nitrate (46 %), sulfate (27 %) and OM (20 %).
As described above, the 21 sites retained for the Limited sites sub-calibration present seasonal,
regional and compositional features consistent with known or expected trends in PM$_{2.5}$ across
the network. The median PM$_{2.5}$ mass at those locations covers a broad range of concentrations
ranging from 0.93 µg/m$^3$ to 13.75 µg/m$^3$ and includes both urban and rural sites. Capturing the
large variability in PM$_{2.5}$ composition and concentration is essential to ensure the proposed site
list is a representative subset of the parent network. However, it should be mentioned that the
proposed site list is not unique but is one of the many feasible solutions since sites whose samples
clustered together in the GMM are likely inter-exchangeable.
### 3.2  Evaluation of Biomass Burning Model
Prior to describing the overall results from the Multilevel model, the Biomass Burning model is
evaluated to determine if the biomass burning model improves predictions for those samples.
To evaluate the quality of the biomass burning model, the predictions are compared to a global
model (section S1) which contains a few samples from all 160 sites which are mostly non-smoke
samples but does contain a few smoke samples.  Visual inspection of Fig. 3 suggests the
equivalence of the biomass burning models to the global model at the lower end of the
concentration range. However, improvement in prediction accuracy can be claimed at high
concentrations for the Biomass Burning model. The gain in model performance is subtle for OC
and TC; however, for EC, predictions benefit from having a dedicated calibration for samples
impacted by wildfire emissions, with an increase in R$^2$ from 0.747 to 0.902 (Section S7).



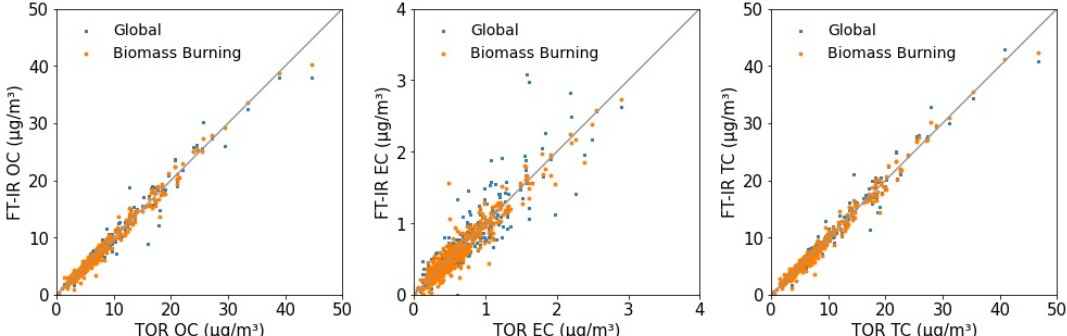

**Figure 3:** Inter-model OC (left), EC (middle), TC (right) comparison between global (section S1)
and Biomass Burning predicted concentrations for the 492 samples classified as biomass burning
in 2015. EC prediction, in particular, benefit from having a dedicated Biomass Burning calibration
model.
Therefore, we retain the biomass burning model as part of the multilevel model and present
the results for the Multilevel model below.
## 3.3    Multilevel modeling – Performance evaluation
### 3.3.1   Carbonaceous aerosol predictions
Figure 4 shows the correspondence between FT-IR Multilevel concentrations for OC and EC and
TOR measurements for 2015 (plot for TC can be found in Section S9) and Table 1 lists the
prediction metrics for all 3 carbonaceous components. The visual agreement between FT-IR
and the reference measurements of OC and EC is high but EC shows higher scatter than the
other measurements. Table 1 indicates that FT-IR OC and TC has higher prediction quality than
EC but both perform satisfactorily. FT-IR OC and TC error is on par with TOR precisions (Table 1)
indicating that FT-IR does not add significant additional error to the measurement. FT-IR EC
predictions, however, have higher error than TOR precision. With respect to reference (TOR)
measurements, concentration-dependent biases in residuals that are determined by the quality
of fit ($R^2$) and dynamic range of the data are expected due to the nature of least-squares
estimation (Besalú et al., 2006; Draper and Smith, 1998). For bias defined as FT-IR predictions
minus the reference (TOR) measurement, least-squares estimator causes an apparent linear
bias which is positive at the low end of the concentration range and negative at the high end of
the concentration range. (see Section S8 for further discussion). The satisfactory agreement
between FT-IR and TOR concentrations as well as the equivalence agreements using the global
model (Section S1) support the validity of the proposed Multilevel modeling in the context of
carbonaceous aerosols prediction from PTFE filters in large speciation networks.

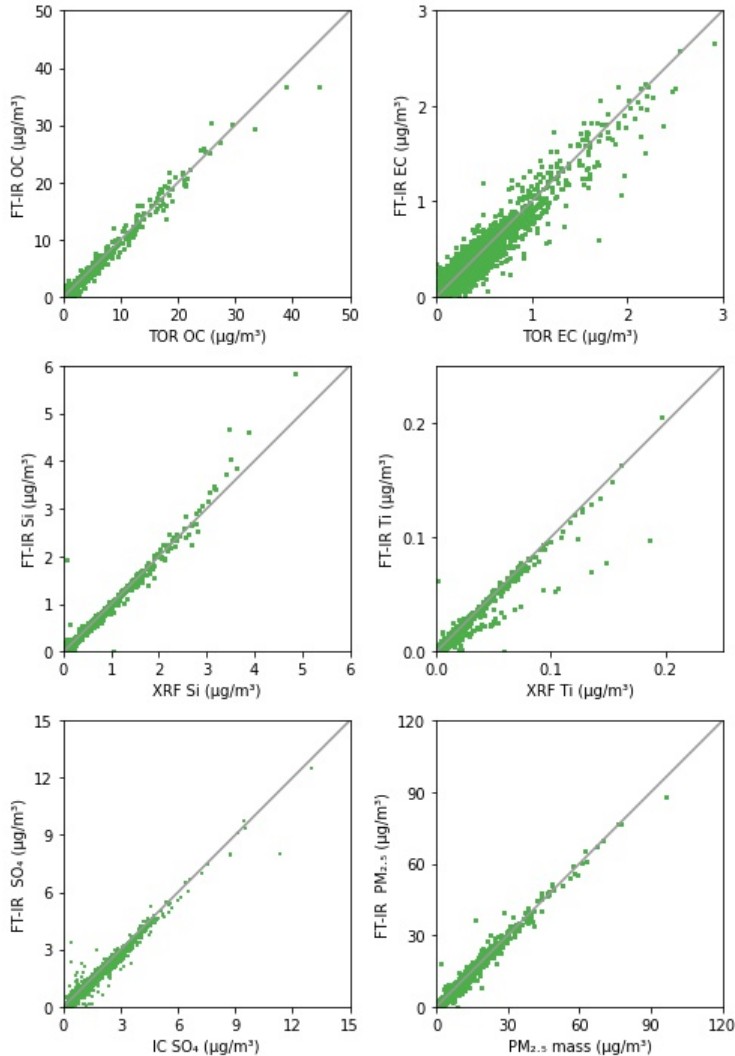

**Figure 4**: Comparison of predicted FT-IR OC, EC, Si, Ti, SO₄ and mass concentrations using the Multilevel model against their reference measurements. Each subplot contains all 19,608 samples collected in the year 2015.

**Table 1**: Summary of Multilevel model performance for IR-active atmospheric constituents for 19,608 spectra analyzed by FT-IR in the year 2015.



| Species | $R^2$ | Bias [µg/m³] | Bias (%) | Error [µg/m³] | Error (%) | Reference Data Error[1] (%) | MDL [µg/m³] | < MDL (%) |
|---|---|---|---|---|---|---|---|---|
| OC | 0.983 | 0.01 | 1.6 | 0.08 | 12 | 10 | 0.06 | 0.9 |
| EC | 0.912 | 0 | 1.7 | 0.02 | 30 | 15 | 0.04 | 20.7 |
| TC | 0.984 | 0.01 | 1.2 | 0.08 | 12 | 11 | 0.07 | 1.3 |
| BC | 0.92 | 0 | -0.3 | 0.03 | 23 | --- | 0.04 | 19.3 |
| Si | 0.983 | 0 | 2.2 | 0.01 | 11 | 13 | 0.01 | 9.7 |
| Al | 0.985 | 0 | 2.2 | 0 | 12 | 10 | 0 | 4.7 |
| Ca | 0.979 | 0 | 1.1 | 0 | 13 | 9 | 0 | 6.9 |
| Ti | 0.941 | 0 | 2.7 | 0 | 21 | 16 | 0 | 14.9 |
| Fe | 0.95 | 0 | 1.1 | 0 | 25 | 8 | 0.01 | 19 |
| SO₄ | 0.983 | 0 | 0.1 | 0.03 | 6 | 2 | 0.03 | 0.9 |
| NO₃ | 0.927 | 0.02 | 15.3 | 0.07 | 54 | 8 | 0.07 | 21.8 |
| PM$_{2.5}$ Mass | 0.985 | 0.03 | 1 | 0.18 | 6 | 6 | 0.25 | 1.1 |

[1]Median relative error for TOR, XRF, IC and gravimetric analysis. OC, EC and TC median relative error estimated
from collocated sampling as measurement error/uncertainty is not reported by IMPROVE for this components.
For all other components, the normalized error was calculated as the uncertainty divided by the concentration
prior to selecting the median. BC is not reported by IMPROVE so measurement error is not estimated.
In addition to OC, EC and TC, light absorption which is predominantly due to black carbon, is also
a measure of one fraction of the carbonaceous aerosol. FT-IR calibrations are found to be
adequate for replicating HIPS BC measurements (Section S9). As expected, the corresponding
model is similar in performance to its EC with $R^2$ and relative error of 0.920 and 23.3 %,
respectively (Table 1). FT-IR BC residuals have a broader interquartile range than in the HIPS BC
collocated data (Section S9). We attribute this effect to a difference in signal to noise ratio and
sensitivity to chemical interferents between the two analytical techniques. While HIPS exploits
the strong absorption properties of refractory carbon in the visible domain, the weak absorptivity
of EC in the mid-infrared domain (Niyogi et al., 2006) and the presence of overlapping species
makes the quantification less accurate.
### 3.3.2  Elemental oxide predictions
Taking advantage of known mineral absorbance bands in the mid-infrared range (Hahn et al.,
2018; Madejová and Komadel, 2001; Senthil Kumar and Rajkumar, 2013) (Section S9), Multilevel
calibrations for soil elements were evaluated for the five crustal elements commonly used to
estimate soil: silicon, aluminum, calcium, titanium, and iron (Table 1 and Fig. 4 for Si and Ti). All
models present a satisfactory agreement between XRF and FT-IR predicted concentrations ($R^2$ >
0.94).  The quality of prediction of the elemental oxides falls into two groups. The first group


includes silicon, aluminum and calcium and is characterized by moderate relative errors (11 – 13
%), similar in magnitude to the FT-IR OC model (12 %) and have similar errors to XRF
measurements indicating similar to OC and TC that FT-IR does not add additional uncertainty.
The second group includes titanium and iron which have larger relative errors (20.9 – 24.8 %),
analogous to HIPS BC and EC models (23.3 – 30 %). Comparing residuals to collocated XRF
measurements (Section S9) shows that the FT-IR based models have a larger interquartile range.
For Fe, XRF uncertainty is quite low and FT-IR adds additional uncertainty to the measurement.
XRF Ti measurements have higher error than the other elements but there is an incremental
increase in error due to FT-IR.  In addition, cross plots of titanium concentrations show a
bifurcation (Fig. 4). While most samples fall near their expected titanium concentration, samples
collected at the Sycamore Canyon (AZ) site present a systematic negative bias, consistent across
years, tentatively attributed to a site-specific soil composition not accounted for by the Limited
calibration. Takahama et al. (2019) demonstrated several methods to identify the possible
occurrence of anomalous predictions in OC and EC based on comparison of new sample spectra
to calibration spectra based on projected PLS scores and regression residual vectors.   These
samples with systematic negative bias in titanium predictions can presumably be identified using
such an approach, provided that compositional differences are detected in the IR spectrum.
Although distinct IR fingerprints exist, FT-IR calibrations for quantifying mineral contents should
be interpreted with care as specific elements may be indirectly quantified through their
correlation with another element even in the absence of clear IR signature (Hahn et al., 2018).
For instance, the variable importance in projection (VIP) scores for the Si, Al, and Ti calibrations
suggests use of similar spectral variables, with small differences, for prediction of these species
(Section S9). However, the 21 GMM sites coverage still meet the necessary requirements for
providing a reliable insight into soil composition in the IMPROVE network.
### 3.3.3   Inorganic ions
The two most abundant inorganic anions quantified in the network: nitrate and sulfate can also
be measured by FT-IR (absorption bands used for prediction are discussed in Section S9). FT-IR
sulfate concentrations display a satisfactory agreement with the reference IR measurements (Fig.
4). Model performance metrics include $R^2$ above 0.98 and relative error lower than 10 % as in the
FT-IR $PM_{2.5}$ model (Table 1). IC measurements of sulfate are very good have even lower error
than FT-IR sulfate.  However, FT-IR nitrate concentrations (Section S9) are characterized by a
moderate drop in the overall model performance ($R^2 = 0.927$) while relative bias and error exceed
15 % and 50 %, respectively and the error far exceeds reference IC nitrate measurement error. A
direct comparison against differential nitrate concentrations at collocated sites highlights the
broad uncertainty in determining nitrate content from PTFE filters (Section S9). Unlike nylon
filters for which nitrate is trapped on the surface, nitrate is known to evaporate from PTFE filters.
This causes a discrepancy between the mass of nitrate deposited onto the nylon filter and the
mass of nitrate on the PTFE filter (Eldred and Ashbaugh, 2004), making FT-IR calibrations with the





nitrate measurements by IC from nylon filters as the reference method error prone.  FT-IR based
nitrate concentrations, measured in this way, should be considered with caution. A possible
alternative is to develop a set of laboratory calibration standards of ammonium nitrate for FT-IR
calibration.  The nitrate mass on the PTFE would be useful for mass closure exercises on the PTFE
filter but would not adequately assess particulate nitrate in the atmosphere.

### 564    3.3.4   PM$_{2.5}$ mass predictions

Since the major aerosol species are shown to be reasonably well measured by FT-IR, it was
anticipated that PM$_{2.5}$ mass calibration would perform well.  The PM$_{2.5}$ model presents reliable
filter mass predictions ($R^2$ = 0.985) characterized by relative bias and error that are 1/3 to 1/2 of
those for OC and on par with gravimetric error (Table 1). The cross plot of gravimetric mass and
FT-IR predictions (Fig. 4) shows that PM$_{2.5}$ mass can be accurately predicted across a broad
concentration range indicating that FT-IR spectra of PTFE filters do not contain interferents or
other limitations that make PM mass predictions error prone.

### 572    3.4    Long term stability

Finally, Multilevel calibrations are extended to 2016 and 2017 to evaluate the inter-year
consistency and determine if the assumptions behind Limited Sites and Biomass Burning models
remain valid over time. For each sampling year, new calibrations were developed following the
framework established for 2015. Models were recalibrated with data from the 21 sites and
biomass burning samples were detected by the functional group screening procedure. Fig. 5
shows the median relative bias (top) and error (bottom) for the three years of data (cross plots
and prediction metrics shown for all predicted species for 2016 and 2017 in Section S10).  These
results indicate that the modeling methodology provided reasonably consistent results across all
three years.
Normalized bias for most species is below 3% and normalized error is consistent for all species
across all three years.  The relative bias for EC and BC are similar to other species in 2015 and
2017 but in 2016 they are larger in magnitude than the other two years and different in sign.
2016 is the first year of TOR data from the multiwavelength TOR instruments (Chow et al., 2015)
so higher bias could be potentially be related the new instruments.  However, the HIPS
instrument was overhauled beginning in 2017 which provides no explanation for high bias in 2016
(http://vista.cira.colostate.edu/improve/Data/QA_QC/Advisory/da0041/da0041_HIPSmodificat
ions.pdf).   Further, the EC and BC calibrations are independent of each other except for using
the same filters for calibrations (as all species do) so the fact that the median bias is roughly equal
but opposite in sign is not due to codependence of the models.
In future work, calibrations models will be updated more frequently than annually with the most
recent year of ambient samples which may smooth biases and errors due to changes to
atmospheric condition and instrument drift.

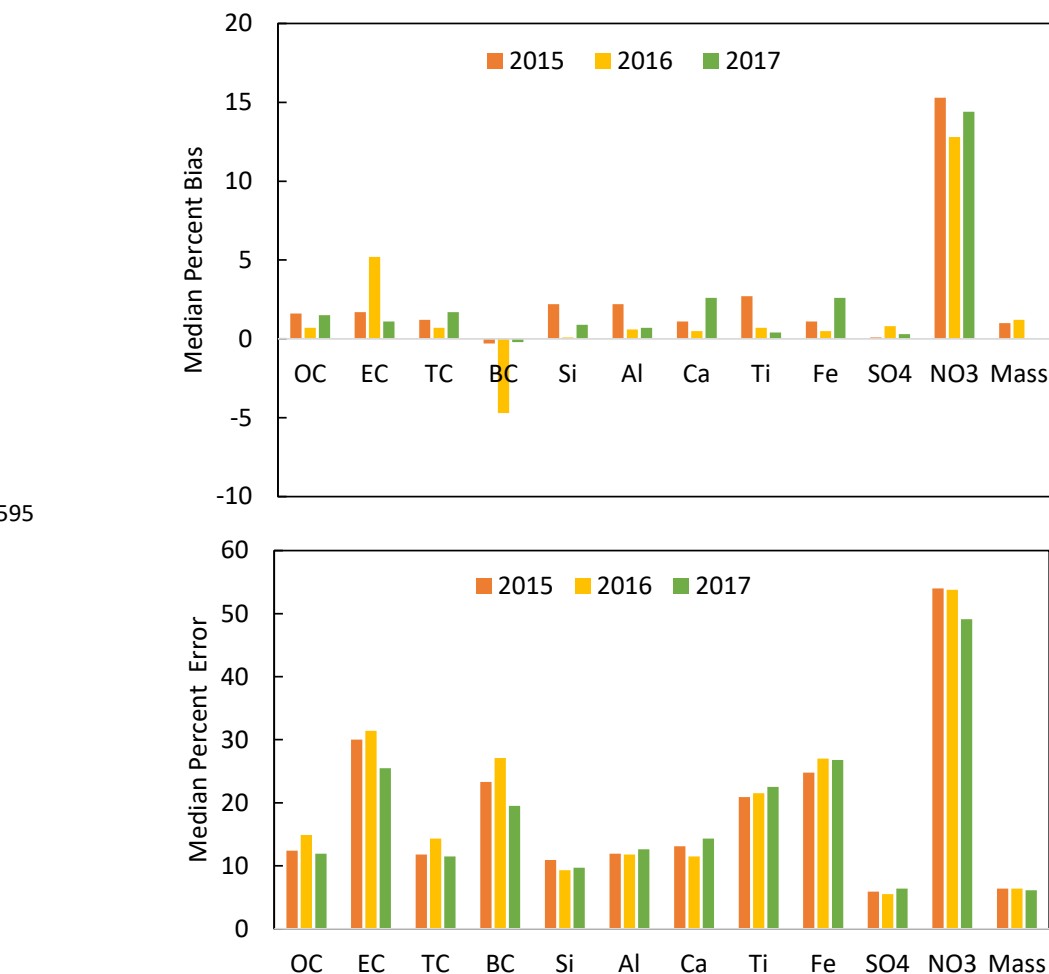



Figure 5. (top) Median Percent Bias and (bottom) Median Percent Error for each constituent
measured for each year.

## 4  Conclusion

In this paper, we investigate the feasibility of an FT-IR method that uses ambient samples as
calibration standards and is adapted for use by a large monitoring network. In this method, all



603 sites in the network collect PTFE filters for FT-IR analysis. A few select sites, used for calibration,
604 would retain all sampling and analyses of current IMPROVE sites to enable re-calibration of the
605 FTIR method on a routine basis. Re-calibration is especially important as atmospheric changes
606 occur and as conditions in the network evolve over time, for example new reference instruments,
607 new or significantly modified FT-IR instruments, changes to sampling protocol or possibly change
608 in filter material. The validity of such a design was evaluated with all PTFE filters collecting $PM_{2.5}$
609 aerosols at 161 IMPROVE sites in 2015 and then tested for all filters in 2016 and 2017.

610 A multi-level modeling algorithm was used whereby smoke impacted samples are identified and
611 predicted by one model and the rest of the samples are predicted by another model developed
612 from 21 selected IMPROVE sites. The data from the two models are combined to evaluate
613 performance of the FT-IR method. The selection of sites was performed such that if one of the
614 21 sites ceases to operate, another site, selected from the same compositional cluster can be
615 used for calibration.

616 The cross-plots and prediction metrics indicate that the Multilevel model is equivalent to
617 conventional calibrations built from samples from every available site. Reliable performance in
618 predicted concentrations were reported for a broad range of atmospheric constituents with
619 detectable infrared signatures such as OC, EC, TC, sulfate, soil elements (Si, Al, Ca, Ti, Fe), light
620 absorption, and $PM_{2.5}$ mass. Due to volatilization off the PTFE filter, nitrate measurements were
621 found to be unsatisfactory. The calibration method was develop using data from 2015, and the
622 same methodology was applied to 2016 and 2017. The model performance metrics in all three
623 years were similar. Results across ~61,500 FT-IR spectra highlight the suitability of the Multilevel
624 calibration design to quantify multiple atmospheric $PM_{2.5}$ species in large monitoring networks.

625 This work presents an alternative, lower cost, filter analysis method to measure speciated aerosol
626 in an operational routine monitoring network. This could be a valuable addition to routine
627 speciated aerosol monitoring networks, such as IMPROVE, by incorporating monitoring sites that
628 collect samples on only a PTFE filter for subsequent analysis. This would provide the opportunity
629 to have a subset of less expensive monitoring site, which could be used for scoping studies to
630 understand the aerosol composition in unmonitored locations. It could also serve as a network
631 cost savings method by having a subset of network sites collect aerosol samples on only a Teflon
632 filter. However, the inability to measure particulate nitrate is a significant deficiency for using
633 this method to replace existing monitoring sites. The FTIR derived aerosol concentrations are
634 also a semi-independent measurement from the routine speciated aerosol measurements.
635 Therefore, routine FTIR measurements would provide valuable QA/QC information for any
636 speciated monitoring network. In addition, FTIR derived concentrations could be used to
637 substitute for missing concentrations in the case where the Teflon sample is valid, but filter
638 samples or analyses on the nylon or quartz fiber filters are not.



For IMPROVE's urban counterpart, the CSN network, after evaluation of the quality of predictions
in CSN, this framework could be used to accomplish goals similar to those of IMPROVE.
Additionally, this method could be used to predict samples collected in the Federal Reference
Method (FRM) network which is a PM mass only network.  Finally, this method, with appropriate
ambient standards, could be applied at other regional or international monitoring networks or
sites to provide low-cost comprehensive composition data.



## 5  Data availability

Data is available at https://doi.org/10.25338/B8TP8V.

## 6  Author contribution

BD developed the software, performed the formal analysis and visualization for much of the manuscript and wrote the original draft of the manuscript, ATW developed software, performed formal analysis and visualization of the GMM work, ST participated in conceptualization, methodology software, visualization and reviewing and editing the manuscript.  KMG developed parts of the biomass burning identification methodology, BS, SC and ASW, provided input throughout the project and reviewed and edited the manuscript , AMD conceptualized of and acquired funding for this project, developed methodology, performed project administration and supervision and reviewed, edited and finalized the manuscript.

## 7  Acknowledgments

The authors acknowledge funding from the National Park Service in cooperation with the Environmental Protection Agency (P18AC01222). Thanks to Anahita Amiri Farahani for assisting with figures.  We are particularly grateful to Kelsey Seibert for overseeing daily FT-IR operations at the University of California Davis and to the numerous undergraduate students who performed spectra collection from 2015 to 2017.

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
