# Peer review of "Quantification of major particulate matter species from a single filter type using"

_Atmospheric Measurement Techniques, 2021_

## Author Comment (AC1)

Review of 'Quantification of major particulate matter species from a single filter type using infrared spectroscopy – Application to a large-scale monitoring network' by Debus et al:

Reviewer #1

Overall, this paper clearly presents a novel solution to increase the efficiency of IMPROVE measurements using FTIR. The results are comprehensive, as well as surprisingly good and consistent, with a few relatively small exceptions.

Thank you for your supportive comments.  Reviewer comments are in black, responses are in green and edits to the paper are in blue.

 In addition to the method comparison, there is a brief discussion characterizing the different sites with their regional similarities, and a slightly awkward overview map showing the CONUS concentrations, but with problematic nitrate.

We hope the map and discussion are helpful.  However, the maps is showing routine IMPROVE data with nitrate coming from IC on nylon filters, not problematic nitrate from PTFE filters.  To clarify, we have modified the caption of the figure as follows.  We have also tried to make the map less awkward by moving the Virgin Islands side to the southeast of the CONUS map and Hawaii to the southwest, more in line with their true geographic locations.

[Figure]

The composition is obtained from routine IMPROVE (non-FT-IR) measurements and the IMPROVE reconstructed fine mass equation (http://vista.cira.colostate.edu/Improve/reconstructed-fine-mass/).

The abstract omits that measurements of nitrate on the PTFE filter cannot adequately access particulate nitrate in the atmosphere. This is a pretty limiting feature and should be highlighted in the abstract, even if it has been reported previously, since it has significant impacts for this method.

Yes, good point. We have modified the abstract by adding the following sentence.

The major limitation is measuring nitrate as it is known to volatilize off of PTFE filters.

The method seems sound in terms of separating training and testing sets, but there are a few aspects that should be shown in more detail:

1)  the outlier results should be shown in supplement.

Thank you for this request. We also thought it would be helpful and so included plots and discussion of the EC outliers in Section S2, page 5 of the supplemental material. The other

samples designated as outliers were excluded prior to evaluation by FT-IR because they are either not in the IMPROVE database or had contamination that was indicated with an SA (sampling anomaly) flag.

2) the comparison before the calibration to the limited sites (at least include in SI).
   We appreciate your interest in these model results. We initially included the global model (which includes all sites) in the main paper but due to the length of the paper and large variety of analyses, we decided to move it to the supplemental material. A brief mention of this model is made in Section 2.5 of the paper and indicates that the results can be found in Section S1 in the supplemental material. In section 3.4, we compare predictions of the biomass burning samples between the biomass burning model and the global model in Figure 3. In section 3.3.1, we mention that the multilevel model preforms similarly well to the global model found in supplemental material.

3) the statistics in the summary table should also be given including outliers and without biomass burning corrections, i.e. an untrained, uncorrected, full-dataset comparison.

Summary statistics for the Global model (which includes all samples) are shown in Table S1-1 in the supplemental material in Section S1.

If these things can be added, the paper would present a much more complete evaluation of the method for future potential users.

Specific Comments

Nitrate is one of the two most abundant inorganic anions that is quantified in the network (if not the most abundant component overall: see Fresno, Ebgert sites) yet determining nitrate from PTFE has a large uncertainty due to volatilization of nitrate on the filter. It was suggested that nylon filters that are analyzed by IC can be used as a reference method but how would this affect costs since this paper promotes FTIR+PTFE as a cost effective single-filter/single-technique.

Thank you for these comments on nitrate, an important and abundant species in aerosols. The efforts presented here utilize IC from nylon filters as the reference method to attempt to measure ambient nitrate which as you noted, but these produce poor results. However, it's useful to have two estimates of nitrate: 1) "true nitrate" in the atmosphere as quantified by the nylon filter and 2) the nitrate on the Teflon filter, which corresponds to its contribution to the gravimetric mass, in principle, is useful as an independent laboratory calibration. We plan to work on this in the future. This cost of this development would be modest. This paper tries to describe the prediction of (1) from the measured signal on the Teflon filter through statistical calibration, which may be using composition particular to season/temperature to correct for the volatilized fraction. While still useful to obtain this estimate via single Teflon filter, there will be limits to how accurate this may be and should be acknowledged as part of reporting, as you suggested. We have added text shown below to address both the limitations of the method proposed in this

paper and address the topic of the value and means for measuring how much nitrate is on the PTFE filter given the volatility issue.

References to papers that successfully performed these alternative techniques for quantifying nitrate would be good to include (for quantifying ammonium nitrate: https://www.tandfonline.com/doi/full/10.1080/02786820701272038).

Thank you for suggesting this and including the reference. We have added the below text and references per your comment.

Unlike nylon filters for which nitrate is trapped on the surface, nitrate is known to evaporate from PTFE filters and extent of volatilization is dependent on temperature and relative humidity during and after sampling. This causes a discrepancy between the mass of nitrate deposited onto the nylon filter and the mass of nitrate on the PTFE filter (Eldred and Ashbaugh, 2004), therefore FT-IR calibrations with the nitrate measurements by IC from nylon filters as the reference should be used with caution. Although there are physical limitations to measuring ambient nitrate on PTFE filters, a measure of nitrate on PTFE filters which corresponds to its contribution to the gravimetric mass is useful for mass closure and data validation. FT-IR has been shown to be useful for measuring and evaluating nitrate under controlled laboratory conditions (ex. Wu et al., 2007). For network samples, nitrate could be measured using laboratory calibration standards and this effort will be addressed in future work.

---

## Author Comment (AC2)

Review of 'Quantification of major particulate matter species from a single filter type using infrared spectroscopy – Application to a large-scale monitoring network' by Debus et al:

Reviewer comments are in black, responses are in green and edits to the paper are in blue.

Reviewer #2

The work by Debus et al. presents a methodology for analysing the aerosol composition using FT-IR analysis of PTFE samples, along with proof-of-concept measurements on samples collected on the USA IMPROVE network. As indicated by the authors, this approach offers a number of advantages over more traditional reference methods for analysing organic and inorganic aerosol composition, being faster and cheaper, making it attractive for large scale monitoring networks.

In my opinion, the authors have done a very good job describing the methodology, from the FT-IR analyses to statistical models used for calibration and prediction, and I was able to follow it even as someone who is non-expert in FT-IR. The reported results demonstrate that the FT-IR was able to predict total mass, organic (OC), elemental (EC) and total (TC) carbon, sulphate, and crustal elements (Si, Al, Ca, Ti, Fe) concentrations with a similar error to reference method. The exception was nitrate which demonstrated higher error, likely due to evaporation from the filter.

Thank you for your comments.  We are pleased to hear that the paper was readable by a non-expert.

My one main comment is I would have liked to have seen more data on how the predicted composition by FT-IR compared to the reference data. My understanding is that 21 sites were used to build the statistical model required to extract the compositional data, and this was then applied to the remaining sites. Fig 4 is the main figure demonstrating how the predicted levels by FT-IR compared to reference for key species but appears to be for all samples and sites in 2015. What I was hoping to see (but may have missed) was similar data/figures demonstrating how the FT-IR predicted composition for individual sites, especially those sites not used to build the multilevel calibration model. This would help I think demonstrate that this method could be applied independently.

Thank you so much for this perspective and comment. We have prepared maps of prediction metrics and reference method concentrations for all predicted species and added a plot (new Figure 5 – included below) in the paper with a few of the metrics for OC, EC, Si and sulfate.  We added maps of the reference method concentrations, FTIR concentrations, and all prediction metrics for all measured constituents in the supplemental material (new section S10 – not included in this response to reviewers but is in the revised supplemental material).  We added text throughout section 3.3 to interpret the maps.

The following is the added text.  Each paragraph is at the end of the section on the give type of constituents.  Following the added text is the new figure 5 referred to in the text.

Annual median maps of FT-IR OC and TOR OC as well as maps of FT-IR EC and TOR EC (Supplemental material S10) are nearly identical.  As shown in Figure 5, annual median OC and EC concentrations are highest at the four urban IMPROVE sites of Seattle, WA, Fresno, CA, Phoenix, AZ and Birmingham, AL than the rural sites and are higher in the east than in the west.  For OC, the relative error is lower than the TOR relative error in the east (where concentrations are higher) and higher than TOR relative error in the west.  OC has an equal or lower number of samples below MDL than TOR at all sites.  For EC, FTIR relative error is higher than TOR relative error at almost all sites.  The percentage of samples of EC that are below MDL for FTIR is similar to are slightly higher than TOR in the eastern US where EC concentrations are higher and are significantly higher than TOR in the western US where concentrations are lower.  These patterns indicate that FTIR does not add error to OC measurements when concentrations are above 0.75 $\mu$g/m$^3$ but does add some error at lower OC concentrations and for EC measurements.

Figure 5 shows the distribution of concentrations of XRF Si across CONUS.  The highest annual median concentrations are in the southwest.  Similar patterns are found for Al, Ca, Ti and Fe except that high Fe concentrations are also observed at the urban sites, particularly Fresno, CA and Birmingham, AL (Figures S10-6 through S10-9).  For Si, FTIR normalized error is lower than XRF in the west where concentrations are higher.  For Ca, Ti and Al, FT-IR normalized error is lower only in the southwest.  For Fe, FT-IR is above XRF normalized error. The percentage of samples below MDL are similar to XRF (0-10% different) in the southwest and central US and modestly higher (15-20%) in the northwest and eastern US for Si.  For Fe, the spatial pattern is similar but the FTIR % below MDL is up to 50 % higher than XRF.  However, for Al, Ca, and Ti, FTIR percent below MDL is approximately the same or lower than XRF at all sites.

The annual median sulfate concentration by IC is shown in Figure 5.  Annual median concentrations are highest in the southeast and eastern US with a gradient in concentrations observed across the midwest.  The median relative error for sulfate by IC is only 2% and all sulfate by FTIR all have higher relative error than 2%.  However, in the eastern US sulfate relative error if less than 15% but in the west, it is considerably higher, peaking in Wyoming where concentrations are very low.   The % below MDL is very similar for FTIR and IC across the continent.  Due to volatility of nitrate, the nitrate metrics for FTIR are not as good as those for sulfate (Figure S10-11).

[Figure]

Figure 5. Annual median reference method concentrations (left), difference in % below MDL (middle) and normalized relative error (right) per site for OC, EC, silicon, and sulfate for CONUS for 2015. For the MDL plot, sites in green and blue indicate that the FTIR has the same of fewer samples below MDL than the reference method. Sites in yellow and red have more samples below MDL for FTIR than for the reference method. For the relative error maps, the median relative error of the reference method estimated using methods described in Table 1 is white. For sites in blue, FTIR has lower relative error than the reference method and sites in red are higher.

Overall, this paper is well written, clearly presented and would be of interest to many in the community.

Thank you for your positive and encouraging comments.

Minor comments

1.  Line 337: Why was 2015 chosen for developing the model, when datasets from later years were available? Would using samples from later help with losses of semi-volatile species?

    All samples were analyzed as soon as possible after being collected so the semi-volatile loses are roughly the same across all years.

2.  Figure 1: Is the composition data presented here measured by FT-IR or the reference methods?

    The data in this plot is from routine IMPROVE measurements.  We have revised a sentence in the figure caption and the first sentence that introduces this figure as follows to clarify this point.

    (caption) The composition is obtained from routine IMPROVE (non-FT-IR) measurements and the IMPROVE reconstructed fine mass equation (http://vista.cira.colostate.edu/Improve/reconstructed-fine-mass/).

    (text) Figure 1 shows the spatial distribution and annual average composition (from routine IMPROVE data) of the 21 sites selected for the Limited Sites model.

3.  Section 3.3.1: As there was good predictive capability for FT-IR for organic carbon, do the authors think that additional information on the organic aerosol could be extracted with FT-IR, perhaps related to the functional groups present?

    Thank you for raising this point.  Yes, we do think that functional group information can be extracted from FT-IR spectra and we have published on this point.   We briefly discuss functional groups in the 4th paragraph of the introduction.  To further this point, we have the below sentence to the end of the paper.

    As shown in our previous work, additional data, including an estimate of organic matter and its functional group composition, can also be obtained from FT-IR spectra of PTFE filters, further increasing the utility of infrared spectroscopy of aerosol samples.

---

## Author Response (AR2)

Review of 'Quantification of major particulate matter species from a single filter type using infrared spectroscopy – Application to a large-scale monitoring network' by Debus et al:

Editors Comments:

Thank you for your responses to the reviewer comments, which you seem to have addressed in full. I have a few minor comments that I would like to see addressed before final publication. First, please specify how the maps in Figures 5 and S10 were built. I cannot find details on how the interpolation between sites was handled. Second, you may want to consider editing Fig 5 so that the site markers are easier to see - even at full zoom the outlines are faint.

Thank you for your comments and for catching my oversight in discussing how the contours were made on the new maps. Below is the text I have added to the manuscript followed by a revised Figure 5 which the site markers in black (instead of gray).

Maps of annual median values of the reference method concentration and performance metric are generated for each aerosol component. Isopleths on the maps were calculated using an ordinary Kriging algorithm which are intended to guide the eye to capture the regional nature of the concentrations and performance quality. For the MDL plot, the difference between the % of samples below MDL for the reference method is subtracted from the % below MDL for FT-IR to indicate if the reference method or FT-IR have more samples below MDL.

[Figure]

Figure 5. Annual median reference method concentrations (left), difference in % below MDL (middle) and normalized relative error (right) per site for OC, EC, silicon, and sulfate for CONUS for 2015. For the MDL plot, sites in green and blue indicate that the FTIR has the same of fewer samples below MDL than the reference method. Sites in yellow and red have more samples below MDL for FTIR than for the reference method. For the relative error maps, the median relative error of the reference method estimated using methods described in Table 1 is white. For sites in blue, FTIR has lower relative error than the reference method and sites in red are higher.